# Learned D-AMP: Principled Neural Network Based Compressive Image Recovery

**Christopher A. Metzler**
Rice University
chris.metzler@rice.edu

**Ali Mousavi**
Rice University
ali.mousavi@rice.edu

**Richard G. Baraniuk**
Rice University
richb@rice.edu

## Abstract

Compressive image recovery is a challenging problem that requires fast and accurate algorithms. Recently, neural networks have been applied to this problem with promising results. By exploiting massively parallel GPU processing architectures and oodles of training data, they can run orders of magnitude faster than existing techniques. However, these methods are largely unprincipled black boxes that are difficult to train and often-times specific to a single measurement matrix.

It was recently demonstrated that iterative sparse-signal-recovery algorithms can be "unrolled" to form interpretable deep networks. Taking inspiration from this work, we develop a novel neural network architecture that mimics the behavior of the denoising-based approximate message passing (D-AMP) algorithm. We call this new network *Learned* D-AMP (LDAMP).

The LDAMP network is easy to train, can be applied to a variety of different measurement matrices, and comes with a state-evolution heuristic that accurately predicts its performance. Most importantly, it outperforms the state-of-the-art BM3D-AMP and NLR-CS algorithms in terms of both accuracy and run time. At high resolutions, and when used with sensing matrices that have fast implementations, LDAMP runs over $50\times$ faster than BM3D-AMP and hundreds of times faster than NLR-CS.

## 1 Introduction

Over the last few decades computational imaging systems have proliferated in a host of different imaging domains, from synthetic aperture radar to functional MRI and CT scanners. The majority of these systems capture linear measurements $y \in \mathbb{R}^m$ of the signal of interest $x \in \mathbb{R}^n$ via $y = \mathbf{A}x + \epsilon$, where $\mathbf{A} \in \mathbb{R}^{m \times n}$ is a measurement matrix and $\epsilon \in \mathbb{R}^m$ is noise.

Given the measurements $y$ and the measurement matrix $A$, a computational imaging system seeks to recover $x$. When $m < n$ this problem is underdetermined, and prior knowledge about $x$ must be used to recovery the signal. This problem is broadly referred to as *compressive sampling* (CS) [1; 2].

There are myriad ways to use priors to recover an image $x$ from compressive measurements. In the following, we briefly describe some of these methods. Note that the ways in which these algorithms use priors span a spectrum; from simple hand-designed models to completely data-driven methods (see Figure 1).

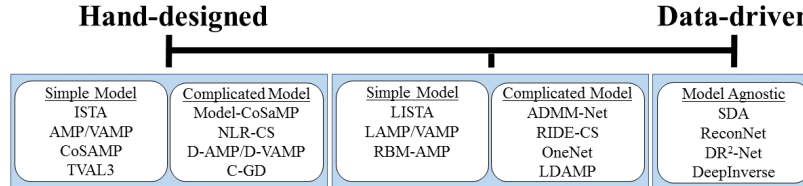

Figure 1: The spectrum of compressive signal recovery algorithms.

## 1.1 Hand-designed recovery methods

The vast majority of CS recovery algorithms can be considered "hand-designed" in the sense that they use some sort of expert knowledge, i.e., prior, about the structure of $x$. The most common signal prior is that $x$ is sparse in some basis. Algorithms using sparsity priors include CoSaMP [3], ISTA [4], approximate message passing (AMP) [5], and VAMP [6], among many others. Researchers have also developed priors and algorithms that more accurately describe the structure of natural images, such as minimal total variation, e.g., TVAL3 [7], markov-tree models on the wavelet coefficients, e.g., Model-CoSaMP [8], and nonlocal self-similarity, e.g., NLR-CS [9]. Off-the-shelf denoising and compression algorithms have also been used to impose priors on the reconstruction, e.g., Denoising-based AMP (D-AMP) [10], D-VAMP [11], and C-GD [12]. When applied to natural images, algorithms using advanced priors outperform simple priors, like wavelet sparsity, by a large margin [10].

The appeal of hand-designed methods is that they are based on interpretable priors and often have well understood behavior. Moreover, when they are set up as convex optimization problems they often have theoretical convergence guarantees. Unfortunately, among the algorithms that use accurate priors on the signal, even the fastest is too slow for many real-time applications [10]. More importantly, these algorithms do not take advantage of potentially available training data. As we will see, this leaves much room for improvement.

## 1.2 Data-driven recovery methods

At the other end of the spectrum are data-driven (often deep learning-based) methods that use no hand-designed models whatsoever. Instead, researchers provide neural networks (NNs) vast amounts of training data, and the networks learn how to best use the structure within the data [13–16].

The first paper to apply this approach was [13], where the authors used stacked denoising autoencoders (SDA) [17] to recover signals from their undersampled measurements. Other papers in this line of work have used either pure convolutional layers (DeepInverse [15]) or a combination of convolutional and fully connected layers (DR$^2$-Net [16] and ReconNet [14]) to build deep learning frameworks capable of solving the CS recovery problem. As demonstrated in [13], these methods can compete with state-of-the-art methods in terms of accuracy while running thousands of times faster. Unfortunately, these methods are held back by the fact that there exists almost no theory governing their performance and that, so far, they must be trained for specific measurement matrices and noise levels.

## 1.3 Mixing hand-designed and data-driven methods for recovery

The third class of recovery algorithms blends data-driven models with hand-designed algorithms. These methods first use expert knowledge to set up a recovery algorithm and then use training data to learn priors within this algorithm. Such methods benefit from the ability to learn more realistic signal priors from the training data, while still maintaining the interpretability and guarantees that made hand-designed methods so appealing. Algorithms of this class can be divided into two subcategories. The first subcategory uses a black box neural network that performs some function within the algorithm, such as the proximal mapping. The second subcategory explicitly unrolls and iterative algorithm and turns it into a deep NN. Following this unrolling, the network can be tuned with training data. Our LDAMP algorithm uses ideas from both these camps.

**Black box neural nets.** The simplest way to use a NN in a principled way to solve the CS problem is to treat it as a black box that performs some function; such as computing a posterior probability.

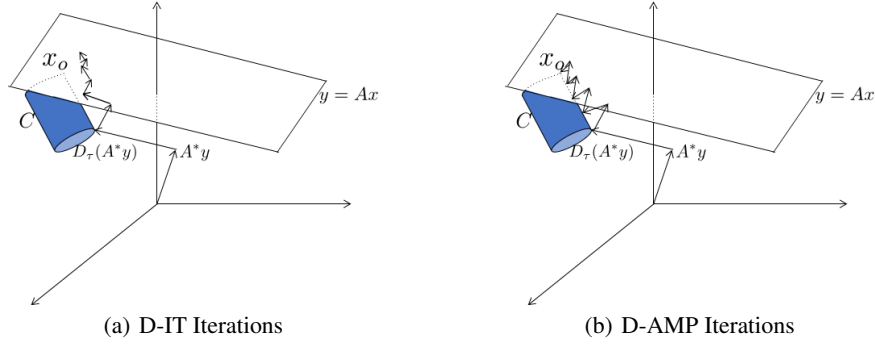

|                          |                          |
|:------------------------:|:------------------------:|
| (a) D-IT Iterations      | (b) D-AMP Iterations     |

Figure 2: Reconstruction behavior of D-IT (left) and D-AMP (right) with an idealized denoiser. Because D-IT allows bias to build up over iterations of the algorithm, its denoiser becomes ineffective at projecting onto the set $C$ of all natural images. The Onsager correction term enables D-AMP to avoid this issue. Figure adapted from [10].

Examples of this approach include RBM-AMP and its generalizations [18–20], which use Restricted Boltzmann Machines to learn non-i.i.d. priors; RIDE-CS [21], which uses the RIDE [22] generative model to compute the probability of a given estimate of the image; and OneNet [23], which uses a NN as a proximal mapping/denoiser.

**Unrolled algorithms.** The second way to use a NN in a principled way to solve the CS problem is to simply take a well-understood iterative recovery algorithm and unroll/unfold it. This method is best illustrated by the the LISTA [24; 25] and LAMP [26] NNs. In these works, the authors simply unroll the iterative ISTA [4] and AMP [5] algorithms, respectively, and then treat parameters of the algorithm as weights to be learned. Following the unrolling, training data can be fed through the network, and stochastic gradient descent can be used to update and optimize its parameters. Unrolling was recently applied to the ADMM algorithm to solve the CS-MRI problem [27]. The resulting network, ADMM-Net, uses training data to learn filters, penalties, simple nonlinearities, and multipliers. Moving beyond CS, the unrolling principle has been applied successfully in speech enhancement [28], non-negative matrix factorization applied to music transcription [29], and beyond. In these applications, unrolling and training significantly improve both the quality and speed of signal reconstruction.

## 2 Learned D-AMP

### 2.1 D-IT and D-AMP

*Learned D-AMP* (LDAMP), is a mixed hand-designed/data-driven compressive signal recovery framework that is builds on the D-AMP algorithm [10]. We describe D-AMP now, as well as the simpler denoising-based iterative thresholding (D-IT) algorithm. For concreteness, but without loss of generality, we focus on image recovery.

A compressive image recovery algorithm solves the ill-posed inverse problem of finding the image $x$ given the low-dimensional measurements $y = \mathbf{A}x$ by exploiting prior information on $x$, such as fact that $x \in C$, where $C$ is the set of all natural images. A natural optimization formulation reads

$$\mathrm{argmin}_x \|y - \mathbf{A}x\|_2^2 \ \text{subject to} \ x \in C. \tag{1}$$

When no measurement noise $\epsilon$ is present, a compressive image recovery algorithm should return the (hopefully unique) image $x_o$ at the intersection of the set $C$ and the affine subspace $\{x|y = \mathbf{A}x\}$ (see Figure 2).

The premise of D-IT and D-AMP is that high-performance image denoisers $D_\sigma$, such as BM3D [30], are high-quality approximate projections onto the set $C$ of natural images.[1,2] That is, suppose

$x_o + \sigma z$ is a noisy observation of a natural image, with $x_o \in C$ and $z \sim N(0, I)$. An ideal denoiser $D_\sigma$ would simply find the point in the set $C$ that is closest to the observation $x_o + \sigma z$

$$D_\sigma(x) = \text{argmin}_x \|x_o + \sigma z - x\|_2^2 \text{ subject to } x \in C. \tag{2}$$

Combining (1) and (2) leads naturally to the D-IT algorithm, presented in (3) and illustrated in Figure 2(a). Starting from $x^0 = 0$, D-IT takes a gradient step towards the $\{x|y = \mathbf{A}x\}$ affine subspace and then applies the denoiser $D_\sigma$ to move to $x^1$ in the set $C$ of natural images . Gradient stepping and denoising is repeated for $t = 1, 2, \ldots$ until convergence.

$$
\begin{array}{lrcl}
\text{D-IT Algorithm} & z^t & = & y - \mathbf{A}x^t, \\
& x^{t+1} & = & D_{\hat{\sigma}^t}(x^t + \mathbf{A}^H z^t).
\end{array}
\tag{3}
$$

Let $\nu^t = x^t + \mathbf{A}^H z^t - x_o$ denote the difference between $x^t + \mathbf{A}^H z^t$ and the true signal $x_o$ at each iteration. $\nu^t$ is known as the *effective noise*. At each iteration, D-IT denoises $x^t + \mathbf{A}^H z^t = x_o + \nu^t$, i.e., the true signal plus the effective noise. Most denoisers are designed to work with $\nu^t$ as additive white Gaussian noise (AWGN). Unfortunately, as D-IT iterates, the denoiser biases the intermediate solutions, and $\nu^t$ soon deviates from AWGN. Consequently, the denoising iterations become less effective [5; 10; 26], and convergence slows.

D-AMP differs from D-IT in that it corrects for the bias in the effective noise at each iteration $t = 0, 1, \ldots$ using an *Onsager correction* term $b^t$.

$$
\begin{array}{lrcl}
\text{D-AMP Algorithm} & b^t & = & \dfrac{z^{t-1}\text{div}D_{\hat{\sigma}^{t-1}}(x^{t-1} + \mathbf{A}^H z^{t-1})}{m}, \\[2ex]
& z^t & = & y - \mathbf{A}x^t + b^t, \\[1ex]
& \hat{\sigma}^t & = & \dfrac{\|z^t\|_2}{\sqrt{m}}, \\[2ex]
& x^{t+1} & = & D_{\hat{\sigma}^t}(x^t + \mathbf{A}^H z^t).
\end{array}
\tag{4}
$$

The Onsager correction term removes the bias from the intermediate solutions so that the effective noise $\nu^t$ follows the AWGN model expected by typical image denoisers. For more information on the Onsager correction, its origins, and its connection to the Thouless-Anderson-Palmer equations [34], see [5] and [35]. Note that $\frac{\|z^t\|_2}{\sqrt{m}}$ serves as a useful and accurate estimate of the standard deviation of $\nu^t$ [36]. Typically, D-AMP algorithms use a Monte-Carlo approximation for the divergence $\text{div}D(\cdot)$, which was first introduced in [37; 10].

## 2.2 Denoising convolutional neural network

NNs have a long history in signal denoising; see, for instance [38]. However, only recently have they begun to significantly outperform established methods like BM3D [30]. In this section we review the recently developed Denoising Convolutional Neural Network (DnCNN) image denoiser [39], which is both more accurate and far faster than competing techniques.

The DnCNN neural network consists of 16 to 20 convolutional layers, organized as follows. The first convolutional layer uses 64 different $3 \times 3 \times c$ filters (where $c$ denotes the number of color channels) and is followed by a rectified linear unit (ReLU) [40]. The next 14 to 18 convolutional layers each use 64 different $3 \times 3 \times 64$ filters which are each followed by batch-normalization [41] and a ReLU. The final convolutional layer uses $c$ separate $3 \times 3 \times 64$ filters to reconstruct the signal. The parameters are learned via residual learning [42].

## 2.3 Unrolling D-IT and D-AMP into networks

The central contribution of this work is to apply the unrolling ideas described in Section 1.3 to D-IT and D-AMP to form the LDIT and LDAMP neural networks. The LDAMP network, presented in (5) and illustrated in Figure 3, consists of 10 AMP layers where each AMP layer contains two denoisers

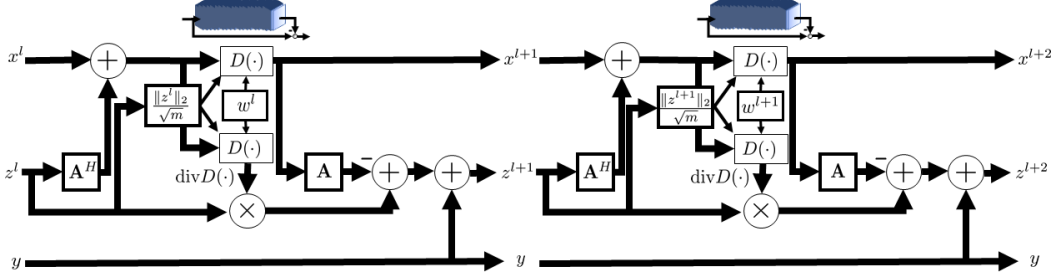

Figure 3: Two layers of the LDAMP neural network. When used with the DnCNN denoiser, each denoiser block is a 16 to 20 convolutional-layer neural network. The forward and backward operators are represented as the matrices $\mathbf{A}$ and $\mathbf{A}^H$; however function handles work as well.

with tied weights. One denoiser is used to update $x^l$, and the other is used to estimate the divergence using the Monte-Carlo approximation from [37; 10]. The LDIT network is nearly identical but does not compute an Onsager correction term and hence, only applies one denoiser per layer. One of the few challenges to unrolling D-IT and D-AMP is that, to enable training, we must use a denoiser that easily propagates gradients; a black box denoiser like BM3D will not work. This restricts us to denoisers such as DnCNN, which, fortunately, offers improved performance.

LDAMP Neural Network

$$
\begin{aligned}
b^l &= \frac{z^{l-1}\mathrm{div}D^l_{w^{l-1}(\hat{\sigma}^{l-1})}(x^{l-1} + \mathbf{A}^H z^{l-1})}{m}, \\
z^l &= y - \mathbf{A}x^l + b^l, \\
\hat{\sigma}^l &= \frac{\|z^l\|_2}{\sqrt{m}}, \\
x^{l+1} &= D^l_{w^l(\hat{\sigma}^l)}(x^l + \mathbf{A}^H z^l).
\end{aligned}
\tag{5}
$$

Within (5), we use the slightly cumbersome notation $D^l_{w^l(\hat{\sigma}^l)}$ to indicate that layer $l$ of the network uses denoiser $D^l$, that this denoiser depends on its weights/biases $w^l$, and that these weights may be a function of the estimated standard deviation of the noise $\hat{\sigma}^l$. During training, the only free parameters we learn are the denoiser weights $w^1, ...w^L$. This is distinct from the LISTA and LAMP networks, where the authors decouple and learn the $\mathbf{A}$ and $\mathbf{A}^H$ matrices used in the network [24; 26].

## 3 Training the LDIT and LDAMP networks

We experimented with three different methods to train the LDIT and LDAMP networks. Here we describe and compare these training methods at a high level; the details are described in Section 5.

- **End-to-end training:** We train all the weights of the network simultaneously. This is the standard method of training a neural network.

- **Layer-by-layer training:** We train a 1 AMP layer network (which itself contains a 16-20 layer denoiser) to recover the signal, fix these weights, add an AMP layer, train the second layer of the resulting 2 layer network to recover the signal, fix these weights, and repeat until we have trained a 10 layer network.

- **Denoiser-by-denoiser training:** We decouple the denoisers from the rest of the network and train each on AWGN denoising problems at different noise levels. During inference, the network uses its estimate of the standard deviation of the noise to select which set of denoiser weights to use. Note that, in selecting which denoiser weights to use, we must discretize the expected range of noise levels; e.g., if $\hat{\sigma} = 35$, then we use the denoiser for noise standard deviations between 20 and 40.

|                      | LDIT | LDAMP |                      | LDIT | LDAMP |
|----------------------|------|-------|----------------------|------|-------|
| End-to-end           | 32.1 | 33.1  | End-to-end           | 8.0  | 18.7  |
| Layer-by-layer       | 26.1 | 33.1  | Layer-by-layer       | -2.6 | 18.7  |
| Denoiser-by-denoiser | 28.0 | 31.6  | Denoiser-by-denoiser | 22.1 | 25.9  |
|                      | (a)  |       |                      | (b)  |       |

Figure 4: Average PSNRs[4] of 100 $40 \times 40$ image reconstructions with i.i.d. Gaussian measurements trained at a sampling rate of $\frac{m}{n} = 0.20$ and tested at sampling rates of $\frac{m}{n} = 0.20$ (a) and $\frac{m}{n} = 0.05$ (b).

**Comparing Training Methods.** Stochastic gradient descent theory suggests that layer-by-layer and denoiser-by-denoiser training should sacrifice performance as compared to end-to-end training [43]. In Section 4.2 we will prove that this is not the case for LDAMP. *For LDAMP, layer-by-layer and denoiser-by-denoiser training are minimum-mean-squared-error (MMSE) optimal.* These theoretical results are born out experimentally in Tables 4(a) and 4(b). Each of the networks tested in this section consists of 10 unrolled DAMP/DIT layers that each contain a 16 layer DnCNN denoiser.

Table 4(a) demonstrates that, as suggested by theory, layer-by-layer training of LDAMP is optimal; additional end-to-end training does not improve the performance of the network. In contrast, the table demonstrates that layer-by-layer training of LDIT, which represents the behavior of a typical neural network, is suboptimal; additional end-to-end training dramatically improves its performance.

Despite the theoretical result the denoiser-by-denoiser training is optimal, Table 4(a) shows that LDAMP trained denoiser-by-denoiser performs slightly worse than the end-to-end and layer-by-layer trained networks. This gap in performance is likely due to the discretization of the noise levels, which is not modeled in our theory. This gap can be reduced by using a finer discretization of the noise levels or by using deeper denoiser networks that can better handle a range of noise levels [39].

In Table 4(b) we report on the performance of the two networks when trained at a one sampling rate and tested at another. LDIT and LDAMP networks trained end-to-end and layer-by-layer at a sampling rate of $\frac{m}{n} = 0.2$ perform poorly when tested at a sampling rate of $\frac{m}{n} = 0.05$. In contrast, the denoiser-by-denoiser trained networks, which were not trained at a specific sampling rate, generalize well to different sampling rates.

# 4 Theoretical analysis of LDAMP

This section makes two theoretical contributions. First, we show that the state-evolution (S.E.), a framework that predicts the performance of AMP/D-AMP, holds for LDAMP as well.[5] Second, we use the S.E. to prove that layer-by-layer and denoiser-by-denoiser training of LDAMP are MMSE optimal.

## 4.1 State-evolution

In the context of LAMP and LDAMP, the S.E. equations predict the intermediate mean squared error (MSE) of the network over each of its layers [26]. Starting from $\theta^0 = \frac{\|x_o\|_2^2}{n}$ the S.E. generates a sequence of numbers through the following iterations:

$$\theta^{l+1}(x_o, \delta, \sigma_\epsilon^2) = \frac{1}{n}\mathbb{E}_\epsilon \|D^l_{w^l(\sigma)}(x_o + \sigma^l \epsilon) - x_o\|_2^2, \qquad (6)$$

where $(\sigma^l)^2 = \frac{1}{\delta}\theta^l(x_o, \delta, \sigma_\epsilon^2) + \sigma_\epsilon^2$, the scalar $\sigma_\epsilon$ is the standard deviation of the measurement noise $\epsilon$, and the expectation is with respect to $\epsilon \sim N(0, I)$. Note that the notation $\theta^{l+1}(x_o, \delta, \sigma_\epsilon^2)$ is used to emphasize that $\theta^l$ may depend on the signal $x_o$, the under-determinacy $\delta$, and the measurement noise.

Let $x^l$ denote the estimate at layer $l$ of LDAMP. Our empirical findings, illustrated in Figure 5, show that the MSE of LDAMP is predicted accurately by the S.E. We formally state our finding.

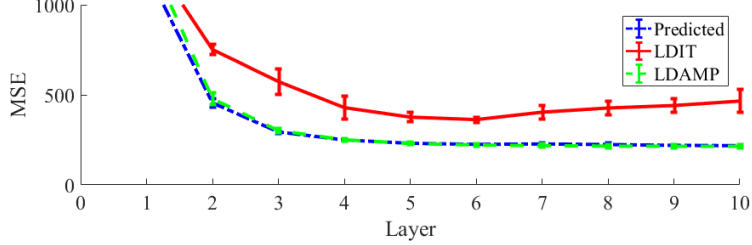

Figure 5: The MSE of intermediate reconstructions of the Boat test image across different layers for the DnCNN variants of LDAMP and LDIT alongside their predicted S.E. The image was sampled with Gaussian measurements at a rate of $\frac{m}{n} = 0.1$. Note that LDAMP is well predicted by the S.E., whereas LDIT is not.

**Finding 1.** *If the LDAMP network starts from $x^0 = 0$, then for large values of $m$ and $n$, the S.E. predicts the mean square error of LDAMP at each layer, i.e., $\theta^l(x_o, \delta, \sigma_\epsilon^2) \approx \frac{1}{n} \left\| x^l - x_o \right\|_2^2$, if the following conditions hold: (i) The elements of the matrix $\mathbf{A}$ are i.i.d. Gaussian (or subgaussian) with mean zero and standard deviation $1/m$. (ii) The noise $w$ is also i.i.d. Gaussian. (iii) The denoisers $D^l$ at each layer are Lipschitz continuous.*[6]

### 4.2 Layer-by-layer and denoiser-by-denoiser training is optimal

The S.E. framework enables us to prove the following results: Layer-by-layer and denoiser-by-denoiser training of LDAMP are MMSE optimal. Both these results rely upon the following lemma.

**Lemma 1.** *Suppose that $D^1, D^2, ...D^L$ are monotone denoisers in the sense that for $l = 1, 2, ...L$ $\inf_{w^l} \mathbb{E} \| D^l_{w^l(\sigma)}(x_o + \sigma\epsilon) - x_o \|_2^2$ is a non-decreasing function of $\sigma$. If the weights $w^1$ of $D^1$ are set to minimize $\mathbb{E}_{x_0}[\theta^1]$ and fixed; and then the weights $w^2$ of $D^2$ are set to minimize $\mathbb{E}_{x_0}[\theta^2]$ and fixed, ... and then the weights $w^L$ of $D^L$ are set to minimize $\mathbb{E}_{x_0}[\theta^L]$, then together they minimize $\mathbb{E}_{x_0}[\theta^L]$.*

Lemma 1 can be derived using the proof technique for Lemma 3 of [10], but with $\theta_l$ replaced by $\mathbb{E}_{x_0}[\theta^l]$ throughout. It leads to the following two results.

**Corollary 1.** *Under the conditions in Lemma 1, layer-by-layer training of LDAMP is MMSE optimal.*

This result follows from Lemma 1 and the equivalence between $\mathbb{E}_{x_0}[\theta^l]$ and $\mathbb{E}_{x_0}[\frac{1}{n}\|x^l - x_o\|_2^2]$.

**Corollary 2.** *Under the conditions in Lemma 1, denoiser-by-denoiser training of LDAMP is MMSE optimal.*

This result follows from Lemma 1 and the equivalence between $\mathbb{E}_{x_0}[\theta^l]$ and $\mathbb{E}_{x_0}[\frac{1}{n}\mathbb{E}_\epsilon \| D^l_{w^l(\sigma)}(x_o + \sigma^l\epsilon) - x_o\|_2^2]$.

## 5 Experiments

**Datasets** Training images were pulled from Berkeley's BSD-500 dataset [46]. From this dataset, we used 400 images for training, 50 for validation, and 50 for testing. For the results presented in Section 3, the training images were cropped, rescaled, flipped, and rotated to form a set of 204,800 overlapping $40 \times 40$ patches. The validation images were cropped to form 1,000 non-overlapping $40 \times 40$ patches. We used 256 non-overlapping $40 \times 40$ patches for test. For the results presented in this section, we used 382,464 $50 \times 50$ patches for training, 6,528 $50 \times 50$ patches for validation, and seven standard test images, illustrated in Figure 6 and rescaled to various resolutions, for test.

**Implementation.** We implemented LDAMP and LDIT, using the DnCNN denoiser [39], in both TensorFlow and MatConvnet [47], which is a toolbox for Matlab. Public implementations of both versions of the algorithm are available at `https://github.com/ricedsp/D-AMP_Toolbox`.

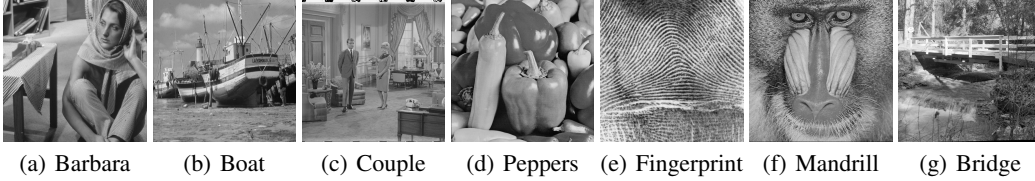

(a) Barbara    (b) Boat    (c) Couple    (d) Peppers    (e) Fingerprint    (f) Mandrill    (g) Bridge

Figure 6: The seven test images.

**Training parameters.** We trained all the networks using the Adam optimizer [48] with a training rate of 0.001, which we dropped to 0.0001 and then 0.00001 when the validation error stopped improving. We used mini-batches of 32 to 256 patches, depending on network size and memory usage. For layer-by-layer and denoiser-by-denoiser training, we used a different randomly generated measurement matrix for each mini-batch. Training generally took between 3 and 5 hours per denoiser on an Nvidia Pascal Titan X. Results in this section are for denoiser-by-denoiser trained networks which consists of 10 unrolled DAMP/DIT layers that each contain a 20 layer DnCNN denoiser.

**Competition.** We compared the performance of LDAMP to three state-of-the-art image recovery algorithms; TVAL3 [7], NLR-CS [9], and BM3D-AMP [10]. We also include a comparison with LDIT to demonstrate the benefits of the Onsager correction term. Our results do not include comparisons with any other NN-based techniques. While many NN-based methods are very specialized and only work for fixed matrices [13–16; 27], the recently proposed OneNet [23] and RIDE-CS [21] methods can be applied more generally. Unfortunately, we were unable to train and test the OneNet code in time for this submission. While RIDE-CS code was available, the implementation requires the measurement matrices to have orthonormalized rows. When tested on matrices without orthonormal rows, RIDE-CS performed significantly worse than the other methods.

**Algorithm parameters.** All algorithms used their default parameters. However, NLR-CS was initialized using 8 iterations of BM3D-AMP, as described in [10]. BM3D-AMP was run for 10 iterations. LDIT and LDAMP used 10 layers. LDIT had its per layer noise standard deviation estimate $\hat{\sigma}$ parameter set to $2\|z^l\|_2/\sqrt{m}$, as was done with D-IT in [10].

**Testing setup.** We tested the algorithms with i.i.d. Gaussian measurements and with measurements from a randomly sampled coded diffraction pattern [49]. The coded diffraction pattern forward operator was formed as a composition of three steps; randomly (uniformly) change the phase, take a 2D FFT, and then randomly (uniformly) subsample. Except for the results in Figure 7, we tested the algorithms with $128 \times 128$ images ($n = 128^2$). We report recovery accuracy in terms of PSNR. We report run times in seconds. Results broken down by image are provided in the supplement.

**Gaussian measurements.** With noise-free Gaussian measurements, the LDAMP network produces the best reconstructions at every sampling rate on every image except Fingerprints, which looks very unlike the natural images the network was trained on. With noise-free Gaussian measurements, LDIT and LDAMP produce reconstructions significantly faster than the competing methods. Note that, despite having to perform twice as many denoising operations, at a sampling rate of $\frac{m}{n} = 0.25$ the LDAMP network is only about $25\%$ slower than LDIT. This indicates that matrix multiplies, not denoising operations, are the dominant source of computation. Average recovery PSNRs and run times are reported in Table 1. With noisy Gaussian measurements, LDAMP uniformly outperformed the other methods; these results can be found in the supplement.

**Coded diffraction measurements.** With noise-free coded diffraction measurements, the LDAMP network again produces the best reconstructions on every image except Fingerprints. With coded diffraction measurements, LDIT and LDAMP produce reconstructions significantly faster than competing methods. Note that because the coded diffraction measurement forward and backward operator can be applied in $O(n \log n)$ operations, denoising becomes the dominant source of computations: LDAMP, which has twice as many denoising operations as LDIT, takes roughly $2\times$ longer to complete. Average recovery PSNRs and run times are reported in Table 2. We end this section with a visual comparison of $512 \times 512$ reconstructions from TVAL3, BM3D-AMP, and LDAMP, presented

Table 1: PSNRs and run times (sec) of $128 \times 128$ reconstructions with i.i.d. Gaussian measurements and no measurement noise at various sampling rates.

| Method | $\frac{m}{n} = 0.10$ | | $\frac{m}{n} = 0.15$ | | $\frac{m}{n} = 0.20$ | | $\frac{m}{n} = 0.25$ | |
|---|---|---|---|---|---|---|---|---|
| | PSNR | Time | PSNR | Time | PSNR | Time | PSNR | Time |
| TVAL3 | 21.5 | 2.2 | 22.8 | 2.9 | 24.0 | 3.6 | 25.0 | 4.3 |
| BM3D-AMP | 23.1 | 4.8 | 25.1 | 4.4 | 26.6 | 4.2 | 27.9 | 4.1 |
| LDIT | 20.1 | **0.3** | 20.7 | **0.4** | 21.1 | **0.4** | 21.7 | **0.5** |
| LDAMP | **23.7** | 0.4 | **25.7** | 0.5 | **27.2** | 0.5 | **28.5** | 0.6 |
| NLR-CS | 23.2 | 85.9 | 25.2 | 104.0 | 26.8 | 124.4 | 28.2 | 146.3 |

Table 2: PSNRs and run times (sec) of $128 \times 128$ reconstructions with coded diffraction measurements and no measurement noise at various sampling rates.

| Method | $\frac{m}{n} = 0.10$ | | $\frac{m}{n} = 0.15$ | | $\frac{m}{n} = 0.20$ | | $\frac{m}{n} = 0.25$ | |
|---|---|---|---|---|---|---|---|---|
| | PSNR | Time | PSNR | Time | PSNR | Time | PSNR | Time |
| TVAL3 | 24.0 | 0.52 | 26.0 | 0.46 | 27.9 | 0.43 | 29.7 | 0.41 |
| BM3D-AMP | 23.8 | 4.55 | 25.7 | 4.29 | 27.5 | 3.67 | 29.1 | 3.40 |
| LDIT | 22.9 | **0.14** | 25.6 | **0.14** | 27.4 | **0.14** | 28.9 | **0.14** |
| LDAMP | **25.3** | 0.26 | **27.4** | 0.26 | **28.9** | 0.27 | **30.5** | 0.26 |
| NLR-CS | 21.6 | 87.82 | 22.8 | 87.43 | 25.1 | 87.18 | 26.4 | 86.87 |

in Figure 7. At high resolutions, the LDAMP reconstructions are incrementally better than those of BM3D-AMP yet computed over $60 \times$ faster.

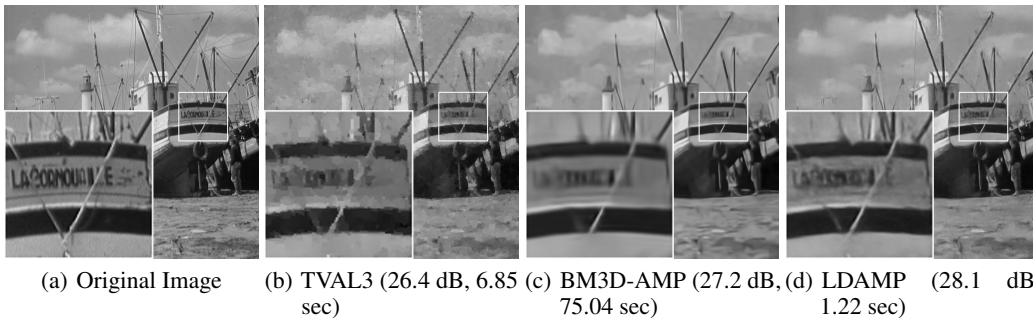

(a) Original Image    (b) TVAL3 (26.4 dB, 6.85 sec)    (c) BM3D-AMP (27.2 dB, 75.04 sec)    (d) LDAMP (28.1 dB, 1.22 sec)

Figure 7: Reconstructions of $512 \times 512$ Boat test image sampled at a rate of $\frac{m}{n} = 0.05$ using coded diffraction pattern measurements and no measurement noise. LDAMP's reconstructions are noticeably cleaner and far faster than the competing methods.

## 6  Conclusions

In this paper, we have developed, analyzed, and validated a novel neural network architecture that mimics the behavior of the powerful D-AMP signal recovery algorithm. The LDAMP network is easy to train, can be applied to a variety of different measurement matrices, and comes with a state-evolution heuristic that accurately predicts its performance. Most importantly, LDAMP outperforms the state-of-the-art BM3D-AMP and NLR-CS algorithms in terms of both accuracy and run time.

LDAMP represents the latest example in a trend towards using training data (and lots of offline computations) to improve the performance of iterative algorithms. The key idea behind this paper is that, rather than training a fairly arbitrary black box to learn to recover signals, we can unroll a conventional iterative algorithm and treat the result as a NN, which produces a network with well-understood behavior, performance guarantees, and predictable shortcomings. It is our hope this paper highlights the benefits of this approach and encourages future research in this direction.

**Acknowledgements**

This work was supported in part by DARPA REVEAL grant HR0011-16-C-0028, DARPA OMNI-SCIENT grant G001534-7500, ONR grant N00014-15-1-2735, ARO grant W911NF-15-1-0316, ONR grant N00014-17-1-2551, and NSF grant CCF-1527501. In addition, C. Metzler was supported in part by the NSF GRFP.

## Footnotes

[1]The notation $D_\sigma$ indicates that the denoiser can be parameterized by the standard deviation of the noise $\sigma$.

[2]Denoisers can also be thought of as a proximal mapping with respect to the negative log likelihood of natural images [31] or as taking a gradient step with respect to the data generating function of natural images [32; 33].

[4]$\mathrm{PSNR} = 10\log_{10}(\frac{255^2}{\mathrm{mean}((\hat{x}-x_o)^2)})$ when the pixel range is 0 to 255.

[5]For D-AMP and LDAMP, the S.E. is entirely observational; no rigorous theory exists. For AMP, the S.E. has been proven asymptotically accurate for i.i.d. Gaussian measurements [44].

[6]A denoiser is said to be $L$-Lipschitz continuous if for every $x_1, x_2 \in C$ we have $\|D(x_1) - D(x_2)\|_2^2 \leq L\|x_1 - x_2\|_2^2$. While we did not find it necessary in practice, weight clipping and gradient norm penalization can be used to ensure Lipschitz continuity of the convolutional denoiser [45].

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
