[Supplementary Material]

# Supplement to "Learned D-AMP: Principled Neural Network based Compressive Image Recovery"

**Christopher A. Metzler**
Rice University
chris.metzler@rice.edu

**Ali Mousavi**
Rice University
ali.mousavi@rice.edu

**Richard G. Baraniuk**
Rice University
richb@rice.edu

Table 1: PSNR of $128 \times 128$ reconstructions with i.i.d. Gaussian measurements and no measurement noise.

| $\frac{m}{n} = .05$ | Barbara | Boat | Fingerprint | Mandrill | Peppers | Couple | Bridge |
|---|---|---|---|---|---|---|---|
| TVAL3 | 19.32 | 21.05 | 16.49 | 21.98 | 19.36 | 21.09 | 19.22 |
| BM3D-AMP | 18.04 | 18.92 | 12.25 | 20.79 | 18.28 | 20.38 | 16.77 |
| LDIT | 17.20 | 19.37 | 16.94 | 20.43 | 17.49 | 19.51 | 18.08 |
| LDAMP | **21.26** | **22.44** | **17.04** | **22.84** | **21.30** | **22.74** | **20.17** |
| NLR-CS | 20.31 | 21.51 | 14.96 | 22.05 | 19.69 | 21.55 | 19.22 |
| $\frac{m}{n} = .10$ | Barbara | Boat | Fingerprint | Mandrill | Peppers | Couple | Bridge |
| TVAL3 | 21.75 | 22.95 | 16.77 | 23.21 | 21.98 | 23.07 | 20.86 |
| BM3D-AMP | 24.36 | 24.04 | **18.07** | 24.12 | 24.44 | 24.79 | 22.05 |
| LDIT | 19.39 | 21.49 | 17.23 | 22.17 | 19.13 | 21.24 | 19.82 |
| LDAMP | **25.80** | **24.67** | 17.36 | **24.37** | **26.00** | **25.43** | **22.52** |
| NLR-CS | 25.17 | 23.84 | 17.84 | 23.56 | 25.18 | 24.91 | 21.80 |
| $\frac{m}{n} = .15$ | Barbara | Boat | Fingerprint | Mandrill | Peppers | Couple | Bridge |
| TVAL3 | 23.47 | 24.25 | 16.97 | 24.09 | 24.18 | 24.74 | 22.00 |
| BM3D-AMP | 27.35 | 25.67 | 19.94 | 25.02 | 27.37 | 26.96 | 23.35 |
| LDIT | 20.53 | 22.54 | 17.29 | 23.17 | 19.74 | 21.89 | 19.50 |
| LDAMP | **28.78** | **26.79** | 17.65 | **25.40** | **29.38** | **27.65** | **23.97** |
| NLR-CS | 28.10 | 25.57 | **20.11** | 24.45 | 28.06 | 26.82 | 23.10 |
| $\frac{m}{n} = .20$ | Barbara | Boat | Fingerprint | Mandrill | Peppers | Couple | Bridge |
| TVAL3 | 25.20 | 25.47 | 17.32 | 24.84 | 25.90 | 26.17 | 23.08 |
| BM3D-AMP | 29.57 | 27.40 | 21.19 | 25.80 | 29.55 | 28.55 | 24.45 |
| LDIT | 20.42 | 22.50 | 17.34 | 22.66 | 21.45 | 22.29 | 20.95 |
| LDAMP | **31.10** | **28.58** | 18.07 | **26.31** | **31.88** | **29.57** | **25.21** |
| NLR-CS | 30.42 | 27.34 | **21.29** | 25.56 | 30.47 | 28.54 | 24.18 |
| $\frac{m}{n} = .25$ | Barbara | Boat | Fingerprint | Mandrill | Peppers | Couple | Bridge |
| TVAL3 | 26.57 | 26.53 | 17.62 | 25.46 | 27.49 | 27.44 | 23.98 |
| BM3D-AMP | 31.39 | 28.89 | 21.97 | 26.50 | 31.27 | 30.01 | 25.37 |
| LDIT | 21.59 | 23.02 | 17.57 | 23.61 | 21.65 | 23.22 | 21.04 |
| LDAMP | **32.68** | **30.17** | 18.13 | **27.06** | **33.87** | **31.06** | **26.20** |
| NLR-CS | 32.42 | 28.78 | **22.04** | 26.52 | 32.47 | 30.13 | 25.26 |

Table 2: PSNR of $128 \times 128$ reconstructions with coded diffraction measurements and no measurement noise.

| $\frac{m}{n} = .05$ | Barbara | Boat | Fingerprint | Mandrill | Peppers | Couple | Bridge |
|---|---|---|---|---|---|---|---|
| TVAL3 | 21.66 | 22.70 | 16.79 | 23.20 | 21.94 | 22.98 | 20.92 |
| BM3D-AMP | 18.37 | 17.89 | 13.46 | 13.32 | 17.80 | 16.91 | 20.83 |
| LDIT | 20.20 | 21.09 | 16.69 | 22.31 | 19.07 | 20.85 | 19.98 |
| LDAMP | **22.10** | **23.45** | **17.33** | **23.54** | **22.32** | **23.50** | **20.94** |
| NLR-CS | 15.14 | 13.44 | 11.73 | 18.43 | 17.23 | 17.70 | 16.04 |
| $\frac{m}{n} = .10$ | Barbara | Boat | Fingerprint | Mandrill | Peppers | Couple | Bridge |
| TVAL3 | 25.08 | 25.39 | 17.34 | 24.79 | 25.83 | 26.13 | 23.13 |
| BM3D-AMP | 25.40 | 24.48 | **18.68** | 24.48 | 25.52 | 25.53 | 22.56 |
| LDIT | 24.02 | 24.77 | 17.31 | 24.74 | 24.36 | 25.24 | 20.21 |
| LDAMP | **27.90** | **26.50** | 17.56 | **25.57** | **28.18** | **27.24** | **23.87** |
| NLR-CS | 24.12 | 21.56 | 17.11 | 21.15 | 24.05 | 22.27 | 21.04 |
| $\frac{m}{n} = .15$ | Barbara | Boat | Fingerprint | Mandrill | Peppers | Couple | Bridge |
| TVAL3 | 27.78 | 27.71 | 17.96 | 26.05 | 28.90 | 28.54 | 24.90 |
| BM3D-AMP | 28.96 | 25.98 | **20.39** | 25.12 | 28.67 | 27.11 | 23.63 |
| LDIT | 28.44 | 26.59 | 17.31 | 25.73 | 30.21 | 26.92 | 24.28 |
| LDAMP | **30.85** | **28.79** | 17.86 | **26.68** | **32.07** | **29.54** | **25.69** |
| NLR-CS | 27.35 | 24.16 | 18.99 | 23.28 | 21.24 | 24.23 | 20.58 |
| $\frac{m}{n} = .20$ | Barbara | Boat | Fingerprint | Mandrill | Peppers | Couple | Bridge |
| TVAL3 | 30.05 | 29.93 | 18.73 | 27.32 | 31.62 | 30.77 | 26.55 |
| BM3D-AMP | 32.08 | 28.07 | **21.41** | 25.89 | 31.47 | 28.99 | 24.57 |
| LDIT | 31.60 | 28.61 | 17.30 | 26.06 | 32.15 | 30.49 | 25.57 |
| LDAMP | **33.10** | **30.20** | 17.94 | **27.59** | **34.06** | **32.17** | **27.24** |
| NLR-CS | 26.41 | 25.26 | 21.04 | 23.48 | 28.18 | 27.67 | 23.47 |
| $\frac{m}{n} = .25$ | Barbara | Boat | Fingerprint | Mandrill | Peppers | Couple | Bridge |
| TVAL3 | 32.37 | 32.13 | 19.63 | 28.63 | 34.20 | 33.05 | 28.12 |
| BM3D-AMP | **34.69** | 29.59 | **22.26** | 26.36 | 34.74 | 30.51 | 25.69 |
| LDIT | 34.00 | 30.43 | 17.65 | 26.24 | 35.37 | 31.02 | 27.63 |
| LDAMP | 34.58 | **32.62** | 18.60 | **29.01** | **36.07** | **33.86** | **28.60** |
| NLR-CS | 31.74 | 28.16 | 20.64 | 25.79 | 28.60 | 28.89 | 21.04 |

Table 3: PSNR of reconstruction of $128 \times 128$ Boat test image with additive white Gaussian measurement noise (AWGN) with various standard deviations (s.d.).

| AWGN with s.d. 10 | | | | | |
|---|---|---|---|---|---|
| **Sampling rate** | $\frac{m}{n} = .05$ | $\frac{m}{n} = .05$ | $\frac{m}{n} = .05$ | $\frac{m}{n} = .05$ | $\frac{m}{n} = .05$ |
| TVAL3 | 21.08 | 22.84 | 24.06 | 24.97 | 25.75 |
| BM3D-AMP | 16.99 | 23.98 | 25.40 | 26.79 | 27.64 |
| LDAMP | **22.41** | **24.56** | **26.36** | **27.62** | **28.56** |
| **AWGN with s.d. 20** | | | | | |
| **Sampling Rate** | $\frac{m}{n} = .05$ | $\frac{m}{n} = .05$ | $\frac{m}{n} = .05$ | $\frac{m}{n} = .05$ | $\frac{m}{n} = .05$ |
| TVAL3 | 21.00 | 22.60 | 23.50 | 23.94 | 24.16 |
| BM3D-AMP | 21.03 | 23.67 | 24.63 | 25.45 | 26.03 |
| LDAMP | **22.33** | **24.24** | **25.35** | **26.33** | **26.75** |
| **AWGN with s.d. 30** | | | | | |
| **Sampling Rate** | $\frac{m}{n} = .05$ | $\frac{m}{n} = .05$ | $\frac{m}{n} = .05$ | $\frac{m}{n} = .05$ | $\frac{m}{n} = .05$ |
| TVAL3 | 20.90 | 22.23 | 22.74 | 22.74 | 22.57 |
| BM3D-AMP | 18.43 | 23.34 | 24.09 | 24.46 | 24.80 |
| LDAMP | **22.11** | **23.80** | **24.55** | **25.10** | **25.34** |
| **AWGN with s.d. 40** | | | | | |
| **Sampling Rate** | $\frac{m}{n} = .05$ | $\frac{m}{n} = .05$ | $\frac{m}{n} = .05$ | $\frac{m}{n} = .05$ | $\frac{m}{n} = .05$ |
| TVAL3 | 20.77 | 21.80 | 21.93 | 21.60 | 21.12 |
| BM3D-AMP | 19.24 | 23.02 | 23.55 | 23.77 | 23.94 |
| LDAMP | **21.85** | **23.41** | **23.90** | **24.33** | **24.50** |
| **AWGN with s.d. 50** | | | | | |
| **Sampling Rate** | $\frac{m}{n} = .05$ | $\frac{m}{n} = .05$ | $\frac{m}{n} = .05$ | $\frac{m}{n} = .05$ | $\frac{m}{n} = .05$ |
| TVAL3 | 20.58 | 21.33 | 21.17 | 20.53 | 19.84 |
| BM3D-AMP | 19.45 | 22.69 | 23.13 | 23.28 | 23.38 |
| LDAMP | **21.58** | **22.96** | **23.44** | **23.65** | **23.76** |