[Reviews · NeurIPS 2017]

Reviewer 1



This is a very interesting work that shows how CNN transform many field BEYOND machine learning, since they are applied here to a reconstruction problem of the compressed sensing/denoising type. The author shows how, using CNN and a trick called "unrolling", they can improve significantly on the Denoising-AMP algorithm. The presented approach outperforms the previous state-of-the-art in terms of both accuracy and runtime. What is interesting is that this further application of the "unrolling trick" illustrate the recent trend towards using training data to improve the performance of iterative algorithms. I do beleive this would be of interest to the NIPS crowd to see the application of these ideas in different dirrection than mainstream machine learning. Minor comments: * I find this sentence ambiguous "First, we demonstrate the state-evolution (S.E.) heuristic, a series of equations that predict the performance of D-AMP, holds for LDAMP as well." It is heuristic or rigorous or demonstrated? The whole "theoretical" section should be a bit more clear about what is computed with heuristic method but is beleived to be exact, what is rigorous, and what is maybe approximative. * As far as I know, AMP has an long history going beyond [5] in statistical mechanics under the name "Thouless-Anderson-Palmer" who introduced the Onsager correction.

Reviewer 2



### Summary In this paper, it is raised a challenge problem in compressive sensing recovery, which is to combine the hand-crafted priors with neural network priors. An unrolled optimization technique is used. The major contribution of this work is to apply the unrolling technique to the D-IT and D-AMP algorithm to form the LDIT and LDAMP algorithm. The improvement in terms of PSNR and time is demonstrated in the Experiments section. Overall, this paper proposes an interesting idea with applications in compressive sensing recovery. However, this paper should benefit from better organization and more clarification of the motivations. - How to demonstrate the deep prior is better, or at least to some extent, than sparse priors? - What is the motivation of using unrolled optimization with deep priors? - Are there any special reasons for the theoretical analysis? Mostly the analysis directly follows from the properties of D-AMP and SGD. A description of how the findings improve the previous results should help the clarification.

Reviewer 3



Summary of the work: The authors present a theoretically motivated NN architecture for solving inverse problems arising in compressed sensing. The network architecture arises from ‘unrolling’ the denoiser-based approximate message passing (D-AMP) algorithm with convolutional denoiser networks inside. The work is nicely presented, and to my knowledge, original. The authors present competitive results on known benchmark data. Overall evaluation: I enjoyed reading this paper, well presented, theoretically motivated work. I was pleased to see that the method achieves very competitive performance in the end both in terms of accuracy and in terms of speed. Comments and questions: 1. Why is the Onsager correction so important?: The Onsager correction is needed because the error distribution differs from the white Gaussian noise assumption typically used to train/derive denoising methods. However, if the whole D-IT unrolled network is trained (or only fine-tuned after initial training) end-to-end, then surely, subsequent denoisers can adopt to the non-Gaussian noise distribution and learn to compensate for the bias themselves. This would mean that the denoisers would no longer be good denoisers when taken out of context, but as part of the network they could co-adapt to compensate for each other’s biases. If this is not the case, why? 2. Validation experiments for layer-wise-training: My understanding is that all experiments use the layer-wise training procedure, and there is no explicit comparison between layer-wise or end-to-end training. Is this correct? 3. Connections to the theory denoising autoencoders: This paper is presented from the perspective of compressive sensing, with solid theory, and unsurprisingly most of the references are to work from this community/domain. However, denoisers are applied more generally in unsupervised learning, and there is a relevant body that the NIPS community will be familiar with which might be worth discussing in this paper. Specifically, (Alain and Bengio, 2012) https://arxiv.org/abs/1211.4246 show that neural networks trained for denoising learn approximate the gradients of the log data generating density. Therefore, one could implement iterative denoising as “taking a gradient step” along the image prior. This is exploited in e.g. (Sonderby et al, 2017) https://arxiv.org/abs/1610.04490 where the connection to AMP is noted - although not discussed in detail. 4. Lipschitz continuity remark: the authors say ConvNets are clearly Lipschitz, otherwise their gradients would explode during training. While this is true, and I would assume doesn’t cause any problems in practice, technically, the Lipschitz constant can change with training, and is unbounded. A similar Lipschitz-continuity requirement arises in - for example - the Wasserstein GAN’s discriminator network, where Lipschitz continuity with a certain constant is ensured by weight clipping. Can the authors comment on the effect of the Lipschitz constant’s magnitude on the findings? Minor comments: Table 1. would benefit from highlighting the fastest/best methods (and those that come within one confidence interval from the winner)